# Fostering Entrepreneurial Ecosystems through the Stimulation and Mentorship of New Entrepreneurs

**Silviu Nate** [1,*] , **Valentin Grecu** [2] , **Andriy Stavytskyy** [3] and **Ganna Kharlamova** [3]

1   Department of International Relations, Political Science and Security Studies, Lucian Blaga University of Sibiu, 550024 Sibiu, Romania
2   Department of Industrial Engineering and Management, Lucian Blaga University of Sibiu, 550024 Sibiu, Romania; valentin.grecu@ulbsibiu.ro
3   Department of Economic Cybernetics, Faculty of Economics, Taras Shevchenko National University of Kyiv, 01601 Kyiv, Ukraine; a.stavytskyy@gmail.com (A.S.); akharlamova@ukr.net (G.K.)
*   Correspondence: silviu.nate@ulbsibiu.ro

**Abstract:** Existing definitions of entrepreneurship highlight the functional role of entrepreneurs, emphasizing their responsibilities for coordination, allocating resources, making decisions, supplying capital, innovation, and bearing uncertainty. This research analyzes the impact that external funding and supportive soft-skills mechanisms such as mentorship, advice, and networking with experienced entrepreneurs have on transforming the entrepreneurial attitude of new entrepreneurs. In measuring attitudes regarding entrepreneurial success, a series of variables specific to the nature of the analyzed entrepreneurial ecosystem are revealed and adapted, starting from psychological research. This approach is implemented to evaluate the self-perception of efficacy and transformation of entrepreneurs after initiating their companies. The survey of Romanian new entrepreneurs is considered as the database. The Global Entrepreneurship Monitor (GEM) data set is used to consider entrepreneurial motives and impacts at the macrolevel. The correlation analysis, statistical tests, and ANOVA helped to reveal the differences in attitudes to mentorship and similar indicators in the Romanian business environment. The novelty of the research is seen in the consideration of field cases and a global monitoring data set through the prism of ground mathematical methods. The focus on boosting new entrepreneurs with a mixture of finance and soft skills support simultaneously addresses a research gap that is slightly closed by this research. The study showed that the mentoring program for new entrepreneurs increased their self-confidence, especially for young people, taught them how to run a company without outside interference, and significantly transformed the mentality of the participants in the experiment. Thus, the policy of supporting new entrepreneurs not only financially, but also in skills, has good prospects and needs to be intensified.

**Keywords:** entrepreneurship; perception of self-efficacy; funding for newly established companies; Romania; GEM; ANOVA; correlation analysis

## 1. Introduction

The concepts of entrepreneurship and economic development and the connection between them have received increasing attention during the past decades [1–4]. Although researchers have not found a general theory of entrepreneurship that is clearly explained in the theory of economic development [5], significant progress has been made in understanding the role of entrepreneurship in economic development [6,7]. The role of entrepreneurship in generating economic growth has been analyzed by policymakers, governments, and international organizations, highlighting that the traditional approach, which argued that large companies were the foundation of a strong economy, is outdated [2,3,5], while small and medium enterprises (SMEs) and new ventures are recognized as important mechanisms for economic development, providing solutions to job creation and increasing per capita income [8]. However, due to their economic strength, large companies have

taken more responsibility for such development, inspiring small and medium-sized enterprises [9] who are now considered to be catalysts of economies, being responsible for innovations that boost economic competitiveness and increase productivity [10–13].

Existing definitions of entrepreneurship underline the functional role of entrepreneurs [14], emphasizing their responsibilities for coordination, allocating resources, making decisions, supplying capital, innovation, and bearing uncertainty [2,15]. Researches show that small and medium-sized enterprises play a crucial role in transforming market economies [3,16], and are considered to be dynamic organic entities [17].

The literature in the field emphasizes many factors that influence entrepreneurial success [18,19]. In addition to several external factors related to current market strategies and government subsidies [20], the entrepreneurial visionary agility and ability to accept and implement change management policies are key variables of entrepreneurial success [21]. The remarkable role is supported by ecosystems that promote for entrepreneurship, comprising the mechanisms, institutions, networks, and cultures that support entrepreneurs [22]. The term entrepreneurial ecosystem emerged only in the 2000s but has become dominant since 2016, while entrepreneurial environment (or similar phrases) was the most common term used in the literature from the 1970s through 2015.

While scientific measurements show that large firms generally have considerable competitive power compared to small enterprises in competitive sectors [23], firms with less than ten employees offer more jobs than large firms [24], which leads to the opinion that finance of start-ups might represent an important factor in supporting economic growth.

The great impact on entrepreneurial success, especially for small enterprises and start-ups, is seen in entrepreneurial self-efficacy [25,26]. The development of the entrepreneurial self-efficacy is seen by most researchers as benefiting from mentorship, advice, facilitation, and coaching [25,27,28].

Based on the taxonomy analyzed by S. Gedeon, attributed, contradictory and diversified connotations were identified to classify the lexicon associated with the word entrepreneurship. However, the fact "that entrepreneurs are as different from one another as they are from non-entrepreneurs" must also be taken into account [29]. Psychological research on entrepreneurship has focused predominantly on personal characteristics as predictors of success, moving beyond the past focus on traits [30]. The present research considers the justification to use variables that refer to awareness and the degree of personal transformation as a key component of entrepreneurial success. Thus, in the study, it seems appropriate to take as a benchmark the Achievement Motivation Inventory (AMI) developed by [31].

The literature shows multiple methods and approaches to analyze the characteristics of entrepreneurs and their impact on venture performance [32–34]. The present research has the goal of analyzing how financial support for establishing new ventures, multiplied by mentorship and similar networking and education factors, can impact new entrepreneurs and their perception of entrepreneurial self-efficacy. For reaching the goal the paper provides an overview of the specific problems and needs of new entrepreneurs funded by the Romania Start-up Plus program and highlights the benefits of such programs in developing entrepreneurial talent and fostering entrepreneurial ecosystems. The structure of the paper is divided into common logical research parts. The literature review describesd and contextualizes the previous and present theoretical background and empirical research on the topic. Then, the methodology part depicts the data set and the survey itself, along with the methods used to assess the current tendencies and validate the conclusions. The methodology part is enhanced by statistical analysis of the current trends in Romanian start-ups and entrepreneurship based on the GEM data, to reveal the microenvironment that affects the new entrepreneurs in Romania. The final part of the paper presents the results of the survey analysis followed by a discussion and conclusions that support the paper with a practical view of the received results.

## 2. Literature Review: Context

Small and medium-sized enterprises (SMEs) are important economic factors that generate jobs, social cohesion, and economic growth. Access to finance is essential to the creation, survival, and growth of SMEs [35], and the global economic and health crises have aggravated the financing constraints of SMEs [36].

These funding challenges are higher for newly established companies. Banks' risk tolerance towards providing credit for new firms has been affected by the economic uncertainty, geopolitical situation, and the last financial crisis [37]. Good credit history and stable financial indicators are demanded by commercial banks to offer loans to companies, and these demands cannot be satisfied by newly established firms [38]. When traditional financing resources, such as loans, overdrafts, or credit lines, are not available, an alternative solution for entrepreneurs to get the necessary funding to start a business is non-traditional financing, such as non-reimbursable grants [37,39].

Governments have taken several measures to support SMEs, especially after the last global economic crisis, to prevent the depletion of their working capital and to enhance their access to financing resources [35]. Access to finance has been acknowledged by G20 leaders as a provider of growth opportunities for companies and economies as a whole and the European Union has financing programs to support the development of new companies [35,36,40].

The Startup Europe Initiative was launched in 2014, as a result of the European Union's acknowledgment of the economic significance of SMEs [41]. This initiative, under the EU Research and Innovation Program - Horizon 2020, aims to foster the development of the European entrepreneurial ecosystem through systemic conditions relating to leadership, networks, talent, and social capital, and through the improvement of institutions and infrastructure [40]. Rossetti et al. [42] present a brief outline of the European startup landscape: "A typical SE beneficiary is an early stage, financially constrained venture that operates in the digital domain and comes from a country with limited private investments in young firms" (ibid., p. 38). As the European Commission believes that "improving the ecosystem for startups and scale-ups in Europe will have a direct beneficial effect on jobs and growth in the EU" [43], another program aimed to support companies that had been recently created or were in their early years of existence [43].

The impact of different factors on the new entrepreneurs is still a question raised in the scientific community [44,45]. Along with the spectrum of successful perspectives for such ventures, most scientists agree on a long list of challenges a new entrepreneur may meet [46]:

1. internal (i.e., developing the business idea and vision, raising capital for start-up, and finding the right business location, funding availability, and accessibility) [47];
2. external (i.e., the form of competition, unforeseen business challenges, the impact of mature entrepreneurs, government and institutional factors, lack of competitiveness, technology innovation and customer loyalty; legal and regulatory framework).

However, the most emergent challenge is seen in the psychological and social issues [48–50], i.e., being optimistic towards the challenges faced, explaining the idea and vision to potential investors, versus lack of planning, skilled labor and proper management skills, etc. Scholars address the latter obstacles a susceptible to policy interventions [51–53], mostly by means of entrepreneurial ecosystems, mentorship, and networks. However, the literature review on the topic revealed that mostly it is researched through the prism of education—students' skills in relation to the issue under discussion [54–57]. On the one hand, the literature shows the value of mentorship in supporting new entrepreneurship, but on the other, it reveals a large research gap regarding other factors to be taken into account.

## 3. Methodology

### 3.1. Entrepreneurial Ecosystem Fostering Program

This study provides an overview of the specific problems and needs of entrepreneurs funded by the Romania Start-up Plus program [58]. It was in order to draw some direc-

tions of action to facilitate a favorable context for the development of the entrepreneurial environment in the Central Development Region of Romania.

The research presented in the paper refers to an entrepreneurial ecosystem, the case of 74 ecosystem startups (which is statistically relevant in sense of sample volume, 95% reliability criteria), which the authors see as part of a broader picture, with a support network that includes entrepreneurs, mentors, employers' associations, investors, and universities [59]. According to the Romania Start-up Plus program [58], the university accessed government funding, organized the process of selecting business plans, and assisted dedicated experts in implementing start-ups, forming a support network that ultimately determined the entrepreneurial ecosystem.

The 74 entrepreneurs who received the minimal aid were asked to complete an online questionnaire, between November and December 2020, to assess the impact that this project had on their business and their individual development.

In measuring the attitude regarding entrepreneurial success, a series of variables specific to the nature of the analyzed entrepreneurial ecosystem were correlated and adapted, starting from psychological research, in order to evaluate the self-perception on efficacy [60] and transformation of entrepreneurs in the implementation of start-ups.

### 3.2. Statistical Analysis of New Entrepreneurship in Romania Case (GEM Report)

Based on the literature review and empirical evidence, the set of hypotheses to access the entrepreneurial environment in Romania can be formulated as:

**Hypothesis 1 (H$_1$).** *There is a difference in the perception of new and experienced entrepreneurs depending on the education environment and economic level of development.*

**Hypothesis 2 (H$_2$).** *The business environment has an expected impact on new entrepreneurship.*

The Global Monitoring of Entrepreneurship (GEM) was selected as a data source. GEM has been collecting data every year since 1998 through adult surveys in each participating country, based on a sample of at least 2000 respondents. The reliability of the data is established using stratified samples from at least 2000 people in the country. The GEM dataset is divided into two main blocks: the Adult Survey (APS) and the National Expert Survey (NES). While the APS data contains data on a diverse number of individual entrepreneurial social and economic characteristics, attitudes, and perceptions of entrepreneurship, NES data explore the social, economic, and political context that shapes the conditions of entrepreneurial activity. The consistency of GEM data is ensured by the fact that the sample is taken from the entire working-age population in each country and thus covers both entrepreneurs and non-entrepreneurs. However, for the purposes and scope of this study on entrepreneurial activity, in particular new entrepreneurship, the focus is on TEA data (general entrepreneurial activity at an early stage).

The methodology of the computational approach to such a massive data set should include a list of methods to support sufficient and reliable results and conclusions, namely: multivariate statistical analysis and descriptive statistical analysis, linear regression, etc. However, the study includes the most appropriate methods for implementing the basic research tasks:

1.  *t*-test (5% verification level) is used to verify the statistical significance of differences;
2.  correlation analysis is used to assess the relationship between business conditions and entrepreneurs' attitudes.

To reflect the new entrepreneurs the TEA indicator is used (Figure 1). As proposed in GEM, TEA is defined as the proportion of adults in the population aged 18 to 64 who are either actively involved in the discovery of a new business or have managed a new business for less than 42 months [61]. Therefore, this definition includes both start-up entrepreneurs and owner-managers of new firms. An individual is considered a "newborn entrepreneur" under three conditions. First, the person has taken steps to create a new business over the

past year. Second, the person expects to share the rights of ownership of a new firm with other persons. Third, the firm has not paid a salary for more than three months. A firm is considered new, however, if salaries have been paid for more than three months, but less than 42 months [62]. The Established Business Ownership Rate, in contrast, represents the percentage of the 18–64 year age-group who are currently an owner-manager of an established business, i.e., owning and managing a running business that has paid salaries, wages, or any other payments to the owners for more than 42 months.

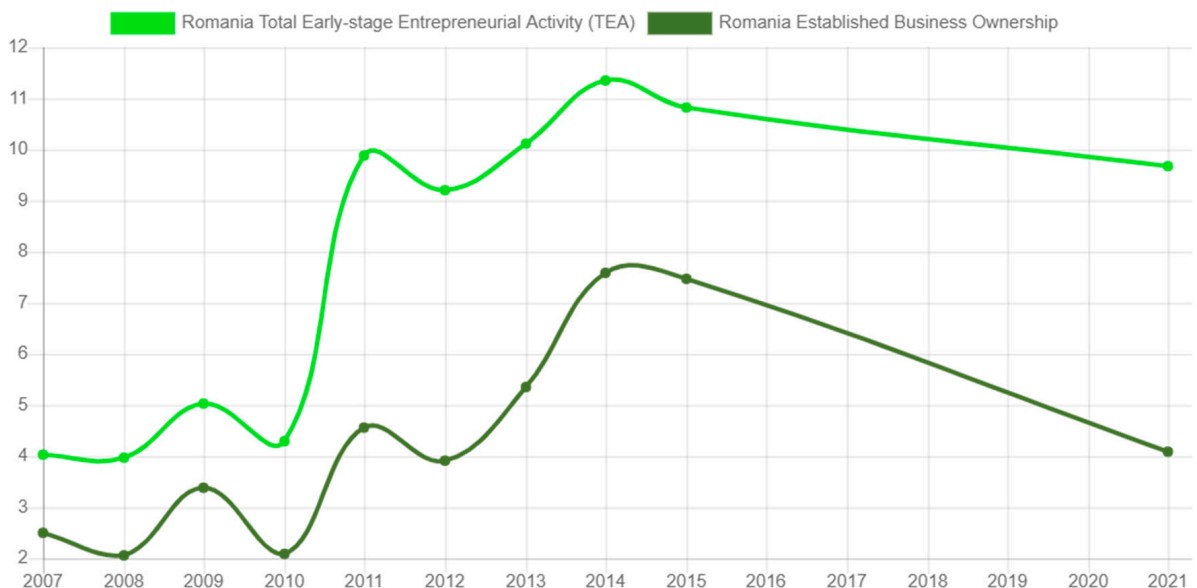

**Figure 1.** The dynamics of new and experienced entrepreneurship in Romania according to GEM report 2021 (APS data). Source: [61,62].

The study seeks to test the above hypotheses on data of more than 2000 (APS) adults (Figure 1), ensuring that it is nationally representative of Romania in 2018 (the latest data edition in GEM). The NES data set is a minimum of 36 carefully chosen experts, who are asked to respond to a series of statements on a Likert scale, rating them from completely false to completely true. The calculations are performed in the EViews soft.

The GEM data set on the APS level permits evaluation of the level of individuals' opinions as to the characteristics, motivations, and ambitions of individuals starting businesses, as well as social attitudes towards entrepreneurship. To consider the environmental factors that could impact the new entrepreneurship in Romania, the following list of APS variables is chosen:

1.  *Perceived Opportunities Rate*: Percentage of 18-64 population (individuals involved in any stage of entrepreneurial activity excluded) who see good opportunities to start a firm in the area where they live;
2.  *Perceived Capabilities Rate*: Percentage of 18-64 population (individuals involved in any stage of entrepreneurial activity excluded) who believe they have the required skills and knowledge to start a business;
3.  *High Status to Successful Entrepreneurs Rate*: Percentage of 18-64 population who agree with the statement that in their country, successful entrepreneurs receive high status;
4.  *Entrepreneurship is a Good Career Choice Rate*: Percentage of 18-64 population who agree with the statement that in their country, most people consider starting a business as a desirable career choice).

The conducted correlation analysis (Appendix A Table A1) claimed all the above factors as highly valued (at 5% statistical significance) for experienced businesses, except the Perceived Opportunities Rate. Meanwhile, new entrepreneurs consider all the above factors as highly valuable and impactful to their decisions to start the business, according to the

correlation analysis (Appendix A Table A1). This is due to the fact that the well established business, first, does not need to wait for new opportunities, and the entrepreneur creates them himself; secondly, it always has a certain margin of safety, even for unsuccessful or partially successful projects, which will not force it to leave the market. At the same time, the failure of the startup project almost automatically precludes its further operation.

According to GEM methodology, there is the option to consider the Romanian entrepreneurial environment through the views of national experts [62]. Thus, the 9 points scoring scale (since 2015) is adopted for the experts' assessments. In other words, GEM now offers all NES quantitative indicators on a 5, 7, and 9 points grade scale. Indeed, the 9 point scales give a more detailed picture of the status of the entrepreneurial framework conditions resulting in more adequate application of sophisticated statistical methods that have requirements about the spread of data, normal behavior, and the like. Thus, the dynamics of the indicators devoted to the paper's topic are presented in Figure 2, which shows experts' views on:

1.  *Governmental Policies (Support and Relevance)*: To which extent the public policies support entrepreneurship as a relevant economic issue;
2.  *Government Entrepreneurship Programs*: The presence and quality of programs directly assisting SMEs at all levels of government (national, regional, municipal);
3.  *Entrepreneurial Education at School Stage*: To which extent the training in creating or managing SMEs is incorporated within the education and training system at primary and secondary levels;
4.  *Entrepreneurial Education at Post-School Stage*: To which extent the training in creating or managing SMEs is incorporated within the education and training system in higher education such as vocational, college, business schools, etc.

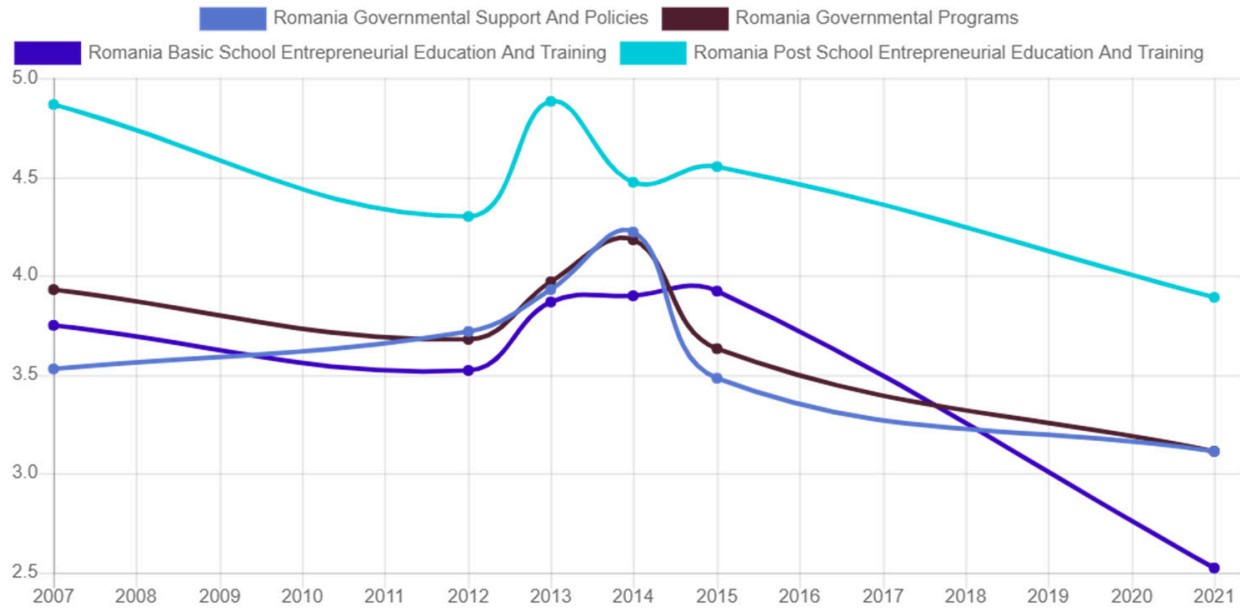

**Figure 2.** The assessment of the Romanian entrepreneurial environment by GEM experts (NES data set). Source: [61,62].

General tendencies in Figure 2 show a decrease in the indicators over time, according to GEM experts.

As the next step of statistical analysis, ANOVA is implemented on the mentioned data set. The statistically significant results as to the impact of the highlighted factors on TEA and experienced business declare:

1.  New entrepreneurs do not have a significant benefit from the considered indicators;
2.  Experienced businesses consider as significant only 2 of them:

a.　　*Commercial and Legal Infrastructure* (i.e., the presence of property rights, commercial, accounting, and other legal and assessment services and institutions that support or promote SMEs);

b.　　*Physical Infrastructure* (i.e., ease of access to physical resources—communication, utilities, transportation, land, or space—at a price that does not discriminate against SMEs).

So, none of the above-stated hypotheses are revealed as valuable for Romanian entrepreneurs in the opinion of the selected GEM experts (NES data set).

*3.3. Survey Fundamentals*

The study uses a tool designed to understand and measure the transformation of the entrepreneurial attitude during the period of assisted implementation of startups, by using variables that refer to the transformative capacity to solve problems by entrepreneurs who initiate startups with the help of funding from external sources.

The variables used in the study refer to the process of entrepreneurial transformation, considering to what extent new entrepreneurs have gained operational experience, their attitudes towards challenges and changes or unforeseen situations, the relationship to the internal business ecosystem and the external competitive environment, the ability to gain autonomy and the desire for a higher status, as well as the level of perseverance and commitment in the medium term.

Starting from the model of predicting entrepreneurial intention advanced by Gorgievski et al. [63], several subscales of entrepreneurial self-efficacy are identified and adapted that aim to clarify the relationship between entrepreneurial self-efficacy and entrepreneurial intention as business continuity vectors of financed startups from government sources. In this regard, 14 items were selected to contain measurement values on the adaptability of entrepreneurs during the startup implementation period.

Another analytical reference model that contributed to the substantiation of the research design is the one advanced by Bradley R. Johnson, which transposes the case of the Achievement Model to entrepreneurs [64]. Thus, the design for the Achievement Questionnaire includes an item scale measuring the extent of agreement or disagreement on variables related to intention, motivational predisposition, interest in starting and growing a business, and ability to predict the direction and limits of the business. All these variables included in the questionnaire give the frame to collect qualitative inputs to appreciate the transformation of startup entrepreneurs in this ecosystem.

The scores on the whole questionnaire and the subscales represent a sum of the points divided by the number of statements. Therefore, the average intensity of the belief about the effectiveness of an entrepreneur for a given variable was calculated. The higher the score, the greater the intensity of the belief. Respondents marked their answers on a 6-degree scale, where 0 indicates "Disagree" and 5 means "Total Agreement". Thus, the qualitative inputs subsequently contributed, through the aggregation and processing of the data obtained, also to quantitative interpretations, to ultimately determining the attitudinal weights that contribute to entrepreneurial success.

*3.4. Sampling Details*

The purposive sampling method is chosen for this study, as this sampling design is based on the judgment of the researchers [65]. Therefore, the sample targeted by applying the online questionnaire consisted of the 74 entrepreneurs, beneficiaries of de minimis aid, in order to conduct comprehensive research to analyze the added value of this project and its contribution to the efforts to develop the entrepreneurial environment in the Central Development Region of Romania.

The research tool used was an online questionnaire, developed on the Google Forms platform, which was distributed to respondents between November and December 2020. The questionnaire includes 17 questions, of which 6 had sub-questions, and 10 classification questions.

All 74 entrepreneurs who obtained financial aid answered the questionnaire. Thus, due to the exhaustive collection of information, this analysis provides a comprehensive picture of the economic situation of the direct beneficiaries of the Start-UP Hub project: Laboratory of Entrepreneurs.

The age distribution (Figure 3a) shows that the majority of de minimis aid recipients are between 30 and 45 years old (over 63%). In terms of gender, a relatively balanced distribution is observed, with a slight advantage for female entrepreneurs (see Figure 3b).

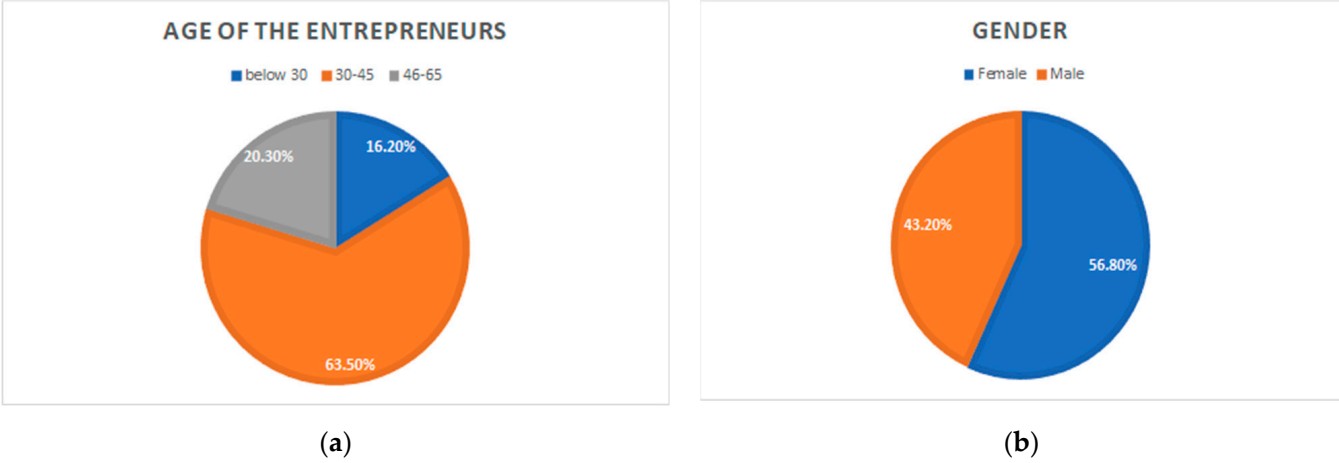

(**a**)                                                                                     (**b**)

**Figure 3.** The demographic representation of the entrepreneurs that participated in the study: (**a**) age of the entrepreneurs; (**b**) gender of the entrepreneurs.

The level of education of entrepreneurs who have submitted business plans chosen for funding under this project is high. Over 90% of them have undergraduate or postgraduate studies, while only 5.4% of respondents have secondary education (see Figure 4).

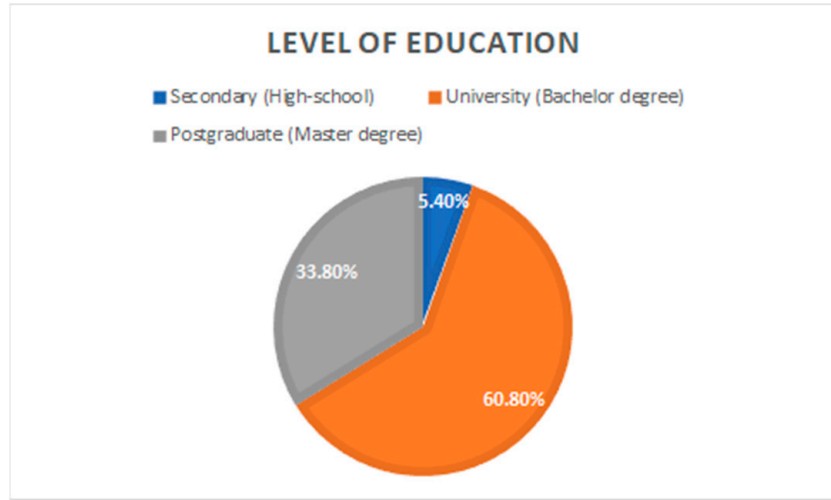

**Figure 4.** The level of education of the respondents.

Data on the main sector of activity of the companies financed within the project, showingtheir absolute and relative numbers, are presented in Table 1. It is clear that the sector of services is best represented among the funded companies, its share in the total being over 70%, of which education services were undertaken by more than 10% of small entrepreneurs, and health services by 13 % of them (Table 1).

**Table 1.** The sector of activity of the newly established companies and gender distribution of the entrepreneurs.

| Object of Activity | Companies | | Female Entrepreneurs | | Male Entrepreneurs | |
|---|---|---|---|---|---|---|
| | **no** | **%** | **no** | **%** | **no** | **%** |
| Manufacturing Industry | 15 | 20.00% | 8 | 18.60% | 7 | 21.88% |
| Construction/building Industry | 3 | 4.00% | 1 | 2.33% | 2 | 6.25% |
| Wholesale and retail trade; repair of motor vehicles and motorcycles | 4 | 5.33% | 2 | 4.65% | 2 | 6.25% |
| Hotels and restaurants | 2 | 2.67% | 0 | 0.00% | 2 | 6.25% |
| Services (consulting, communications, IT&C, advertising, photography, rental, tourism, cultural-entertainment, repairs, beauty) | 32 | 44.00% | 16 | 39.53% | 16 | 50.00% |
| Education | 8 | 10.67% | 7 | 16.28% | 1 | 3.13% |
| Healthcare and social work | 10 | 13.33% | 8 | 18.60% | 2 | 6.25% |
| Total | 74 | 100% | 42 | 100% | 32 | 100% |

Analyzing the businesses in the field of education, one can notice that they were made up mainly of women entrepreneurs (87.5%) over the age of 30, and 80% of the activities in the field of healthcare were initiated by women entrepreneurs. On the other hand, both the firms operating in the field of hotels and restaurants have been set up by male entrepreneurs up to the age of 45. The 74 implemented projects covered a wide variety of types of economic activities, these being also the expression of a very diverse panel of entrepreneurs in terms of age, level of education, managerial experience, etc. Businesses in the field of services, regardless of their nature, represent three-quarters of the total activities established, of which only 9 businesses are in the area of services addressed to corporate clients (B2B), and 47 businesses operate in the area of services to the population.

Businesses in the service sector, with a high degree of homogeneity, include:

1. Sports equipment rental services or services addressed to the population traveling for tourism purposes (related to tourism), such as the rental of sports equipment (7 supported enterprises—9.5% of the total)
2. Other forms of education such as after-school or various courses for children and pupils (7 supported enterprises—9.5% of the total)
3. Dental services (6 supported companies—8% of the total)
4. Construction, maintenance, and interior design services (4 supported enterprises—5.5% of the total)

The project financed 12 business plans for companies that are active in the productive/industrial sector, including various activities such as the manufacture of toys, furniture, advertising materials, bread/pastry/confectionery (2 businesses), tools, educational books in textiles, textile/garments industry, crimping hydraulic hoses or the production of 3D printed plastic elements. Among 12 businesses, 7 were established by males and 5 by females. Regarding the age of entrepreneurs with business in the industrial/productive sector, 4 of them are young people under 30 (of whom 3 are men), 7 entrepreneurs are between 30 and 45 years old and only one entrepreneur is over 45 years old (a woman). A specific characteristic of entrepreneurs with business in the production area is the level of education, where 5 entrepreneurs have higher education (42%), another 5 entrepreneurs (42%) have postgraduate studies and only 2 people have high school education (16%).

## 4. Results

Regarding the transformation of the entrepreneurial attitude, the answers of the respondents are analyzed based on their age, education, gender, and previous experience in the field of their new business. The results show to what extent the entrepreneurs have adapted to the challenges encountered following the implementation of the business plan and after finalizing the sustainability period (18 months after establishing the new firm). The 74 entrepreneurs were required to answer the questions presented in Table 2.

**Table 2.** Transformation of perception on self-efficacy of the entrepreneurs who benefited from financial aid to start their company and from mentorship to run their business in the first 18 months.

| To What Extent Do You Agree with the Following Statements? | Totally Disagree | Disagree | Neutral | Agree | Totally Agree |
|---|---|---|---|---|---|
| I have become a more flexible person; I have the willingness to act in new situations and change things that do not work optimally. | 2.70% | 4.05% | 22.97% | 47.30% | 22.97% |
| I overcame my fear of failure. | 5.41% | 9.46% | 35.14% | 32.43% | 17.57% |
| I take on difficult tasks, better-set requirements in the business, and better appreciate the risks. | 2.70% | 5.41% | 27.03% | 36.49% | 28.38% |
| I became more independent and self-confident. | 4.05% | 2.70% | 25.68% | 43.24% | 24.32% |
| I have much greater confidence in my success from now on. | 4.05% | 5.41% | 27.03% | 39.19% | 24.32% |
| I discovered that I like to lead and enter business competitions. | 6.76% | 13.51% | 24.32% | 35.14% | 20.27% |
| Implementing this business has helped me realize that I need to constantly improve and learn new things. | 2.70% | 8.11% | 10.81% | 22.97% | 55.41% |
| From now on I can set goals much more clearly and pertinently. | 2.70% | 8.11% | 18.92% | 39.19% | 31.08% |
| I discovered what compensatory effort means. We overcame fear by discovering latent personal resources that came to the surface through sustained involvement. | 12.16% | 10.81% | 29.73% | 29.73% | 17.57% |
| I want more and aspire to a higher status as an entrepreneur. | 1.35% | 10.81% | 13.51% | 29.73% | 44.59% |
| I am proud of the productivity of my effort. | 2.70% | 5.41% | 18.92% | 31.08% | 41.89% |
| Now I am making stronger commitments and I have become a much more responsible person and possess increased inner autonomy. | 4.05% | 9.46% | 18.92% | 39.19% | 28.38% |
| I can focus much better on tasks and problems without being distracted by other disruptive factors. | 5.41% | 14.86% | 21.62% | 39.19% | 18.92% |
| I have become more persistent; I have more endurance and strength to perform tasks and delegate tasks to others. | 2.70% | 9.46% | 21.62% | 39.19% | 27.03% |
| I discovered that I like to lead and enter business competitions. | 6.76% | 13.51% | 24.32% | 35.14% | 20.27% |
| Implementing this business has helped me realize that I need to constantly improve and learn new things. | 2.70% | 8.11% | 10.81% | 22.97% | 55.41% |

The respondents declared that the experience of starting a business and receiving advice from mentors had made them more flexible and more willing to act in new situations and change things that do not work optimally. 73% of entrepreneurs with university degrees and 68% of those with postgraduate studies appreciate to a large and very large extent that they have developed the ability to act in new situations. Women find greater adaptability following the implementation of the business plan (76.2%), while 62.5% of male entrepreneurs say they have become more flexible and more adaptable. There is an average positive perception of 71.6% of both genders who appreciate to a large and very large extent an adaptation to new situations, and a positive perception of personal and entrepreneurial transformation.

Regarding age categories, 75% of the age group under 30 years, 66% of the age category 30–45 years, and 80% of the age category 46–65 years appreciate to a large and very large extent the capacity for flexibility and adaptation to new situations. While entrepreneurs under the age of 30 are not found in the area of negative values, in the age segment of 30–65 we find an average of 27% which shows a poor evolution and some entrepreneurial rigidity in new situations (Figure 5).

Entrepreneurs with no practical experience or less than 1 year of management experience, at the time of accessing the financing, indicate an average gain of 95% in terms of flexibility and adaptation to new situations, while entrepreneurs with experience between 1–3 years and over 3 years indicate an average positive change of 93.05%. This confirms the hypothesis that the implementation of a startup helps entrepreneurs to become more flexible and to adapt more easily to new situations.

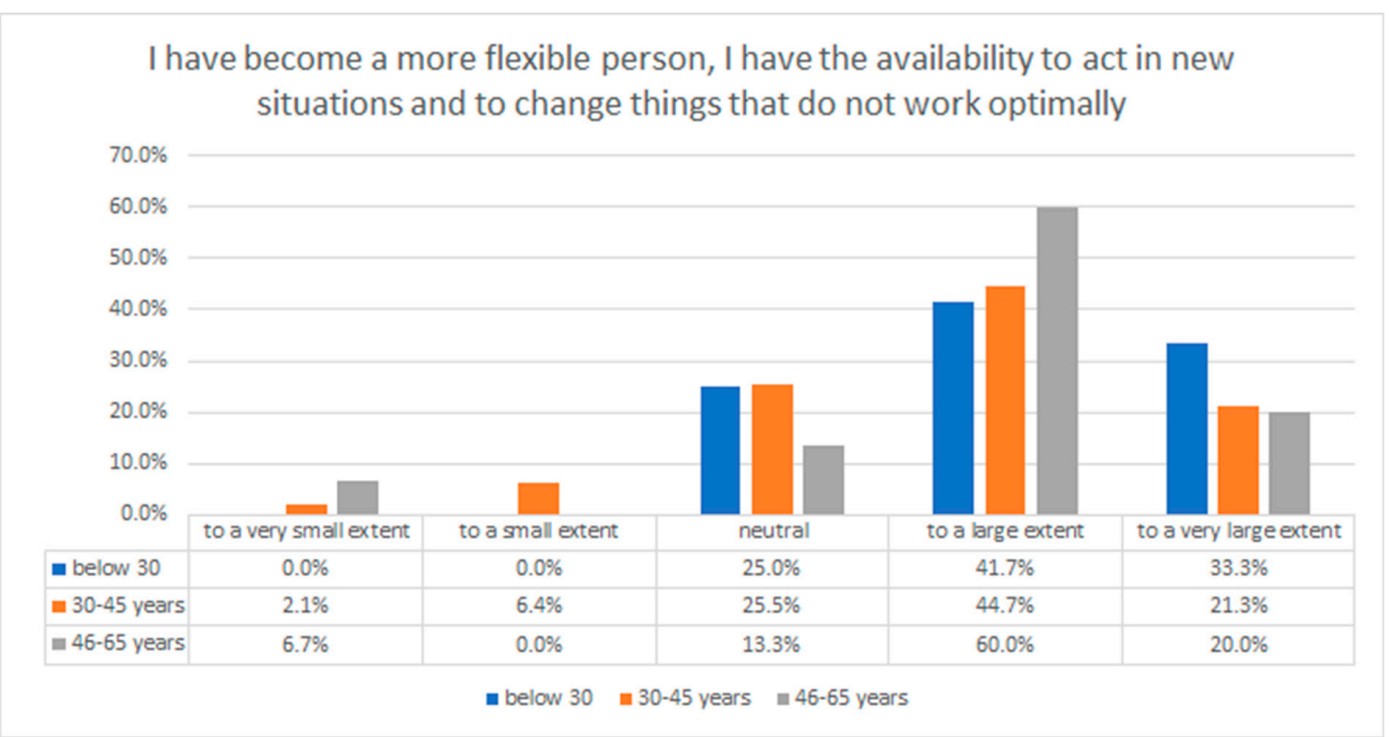

**Figure 5.** The flexibility of entrepreneurs by age categories.

54.8% of female respondents consider that they overcame their fear of failure, while 40.5% consider that they did not overcome this fear, nor did it represent a major obstacle in the design and development of the business. Regarding the category of male entrepreneurs, 43.8% consider that they overcame their fear of failure, and 28.1% consider that they did not overcome this fear, nor did it represent a major obstacle in the design and development business. Thus, one can observe a higher level of entrepreneurial anxiety or an increased perception of risks in men, to a large and very large extent of 28.1%, while only 4.8% of female entrepreneurs indicated the perception of fears until the end of the sustainability period.

However, there is also an appreciable percentage of uncertainty by age categories: 41.7% of entrepreneurs under 30 could not identify to what extent they overcame this fear, 36.2% in the age category 30–45 years, and 26.7% of the age category 46–65 years.

Nevertheless, the cumulative positive and neutral average at all levels of entrepreneurial experience in overcoming the fear of failure stands at 88.77%.

Concerning the willingness to take on difficult tasks, the ability to better set requirements in the business, and better appreciate the risks, female respondents developed higher entrepreneurial confidence, with 69% considering that they had acquired a large and very high capacity to run the business, while men's responses reflected that 59.4% of them had reached the threshold of managerial efficiency.

The age group 30–45 years is the most confident in acquiring these skills, with a proportion of 70.2%, while the age groups under 30 years and those between 46–65 years have average confidence, at 55.8%.

Approximately 31% of all respondents could not identify an evolution or involution of entrepreneurial skills. The analysis of this item confirms the hypothesis that the implementation of a business helps entrepreneurs to prioritize actions and better anticipate unforeseen situations.

The hypothesis that implementing a business increases the self-confidence of entrepreneurs and makes them more independent has been investigated. 73.8% of female entrepreneurs consider that they have gained self-confidence following the implementation of the business plan, compared to 59.4% of male entrepreneurs.

The extremes regarding the perception of personal development and the increase of self-confidence can be identified in the age category under 30 years with a satisfaction level of 91.7% and a more weighted confidence level of 53.3% for the respondents aged 46 to 65 years.

Consequently, it can be concluded that in people under 30 the enthusiasm is higher so that the level of satisfaction they perceive after the successful implementation of the business plan is higher than in the age category over 45 years.

Respondents with postgraduate studies are much more determined and confident about personal success in future business. Thus, 72% of them consider that they have gained much more confidence for their success from now on, while other respondents with higher education are about 10 percent less confident, and only 62.2% believe that they have gained the ability to have entrepreneurial success from now on.

Regardless of the experience prior to the implementation of the startup, an average of 68.2% of the beneficiaries of the de minimis scheme have high and very high confidence in their success from now on (Figure 6).

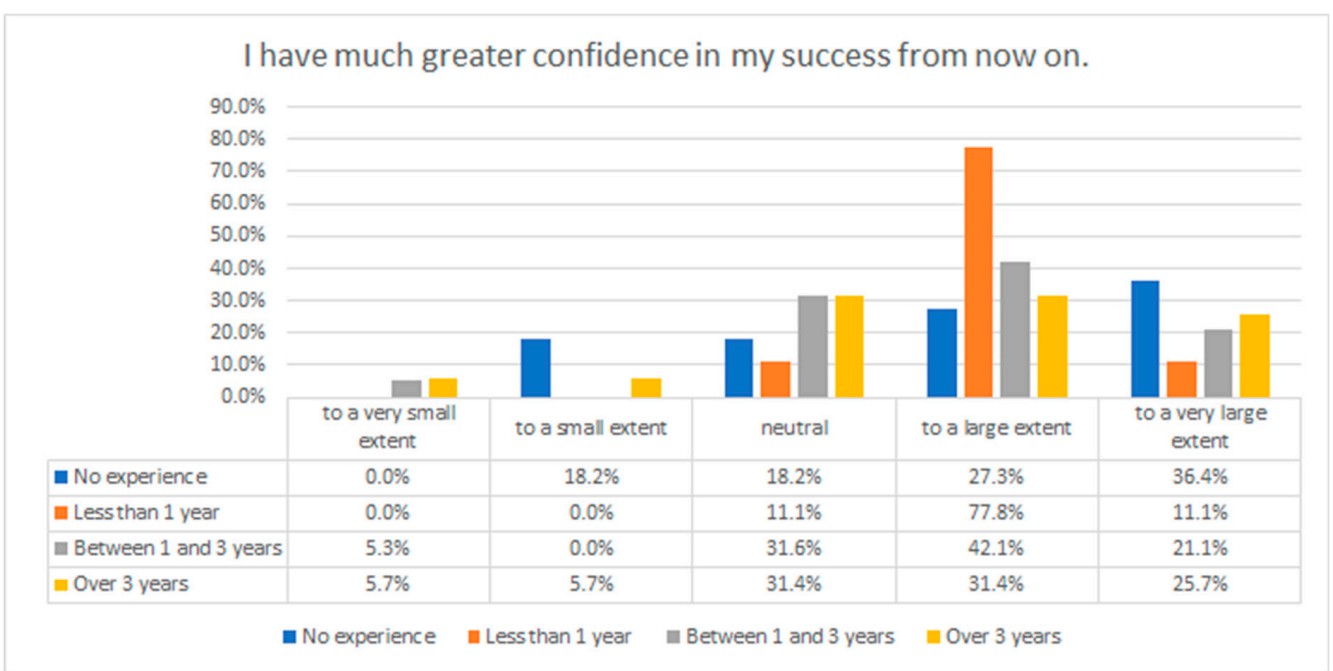

**Figure 6.** Confidence in entrepreneurial skills based on previous experience.

Participation in the project that offered entrepreneurs not only financial support but training, monitoring, and constant support from experts of the implementation team confirms the appropriate framework for developing the skills to become a good entrepreneur.

For 62.5% of male entrepreneurs and 50% of female entrepreneurs, it is confirmed that the implementation of a business stimulates competitive spirit and leadership. However, 25% of male entrepreneurs and 16.7% of female entrepreneurs do not find themselves in/or are not accustomed to the competitive business environment. The most difficult entrepreneurial path seems to be in the segment of respondents with no previous practical experience; only 45.5% of them appreciate to a large and very large extent that they like to drive and enter business competitions. In conclusion, the experience can be crucial for engaging in a competitive spirit and taking on the challenges of running a business, while implementing a business stimulates the competitive spirit and leadership.

Implementing a business helped the respondents realize that they need constantly to improve and learn new things. 83.3% of female entrepreneurs noted that they need personal upgrading continuously. Male respondents were less sure of the latter (71.9%). All entrepreneurs with previous experience under 1 year (100%) believe that the imple-

mentation of this business has helped them to realize that they need to constantly improve and learn new things. Therefore, it can be concluded that the implementation of a business leads to an objective awareness of the need for personal development and improvement/professionalization. This fully correlates with previous studies [66–68].

The experience of starting and running a business allows entrepreneurs to set goals much more clearly and pertinently. There is a value of 73.8% among female entrepreneurs, with high and very high confidence in their ability to have entrepreneurial coherence and high and very high confidence of 65.6% in male entrepreneurs.

86.7% of respondents aged 46–65 years compared to 66% of respondents aged under 30 and 30–45 have high and very high confidence in their ability to set goals more clearly and pertinently.

Although male entrepreneurs have less confidence in their entrepreneurial success (10% less than female entrepreneurs), less appetite for leadership (12.5% less than female entrepreneurs), and less self-confidence (14.4% less than female entrepreneurs), they aspire to a greater extent (78.1%) than female respondents (71.4%) to a higher status as an entrepreneur.

There is a difference in perception at the level of gender categories, generated by the fact that men have a lower fear of failure than women (43.8% vs. 54.8%) and have less confidence in their entrepreneurial success, but have higher expectations regarding the evolution of the business which indicates a greater need for success among men or a more proactive behavior in women.

The level of entrepreneurial satisfaction among female respondents (83.3%) is 23.9% higher than that of male respondents (59.4%). The overall average entrepreneurial satisfaction, regardless of age group, is 73.5%, showing that three-quarters of entrepreneurs are proud of the productivity of the efforts made.

Three times more men (21.9%) than women entrepreneurs (7.1%) believe that they have failed to increase their level of personal autonomy. However, 73.8% of female respondents and 59.4% of male respondents believe to a large and very large extent that the implementation of the business has helped them to make firmer commitments and become more responsible people. A cross-sectional average of 69.2%, calculated for all age groups, considers that they have significantly or decisively increased their level of personal autonomy.

In order to approve the statistical verification, the results received were tested by one-factor analysis (ANOVA). The probability of less than 0.05 revealed the statistical difference in the results in Table 1 (F-test = 56.13). Hence, there is indeed a difference in responses depending on the considered attributes of the entrepreneurs' sample.

## 5. Discussion

Implementing a business has multiple implications on a personal and professional level, and the 74 entrepreneurs funded in the Start-UP Hub project appreciate that at the end of the sustainability period they acquired skills that helped them improve their managerial performance. Thus, the ability to adapt was mentioned by 77.03% of respondents as a skill acquired in this project, followed by a positive attitude in problem-solving and communication skills. Additional to developing managerial abilities, entrepreneurs can rely on custom apps and software solutions to help them manage ordinary and extraordinary challenges [69].

Of course, this study has its limitations, in particular, due to the sample size. In fact, all 100% of respondents were interviewed, but the total value of 74 does not seem to be completely sufficient to form general conclusions and ensure that the results are unbiased. Therefore, to confirm the further policy recommendations, it is necessary to repeat this analysis either with another broad sample or for another country.

However, this analysis is limited by several factors that influenced it. The purposive sampling method is applied [65]; thus, the statistics presented in this paper are predominantly descriptive. The involvement of the respondents when completing the question-

naires is varied, and among the causes, we mention a large number of questions, the time of year when the data were collected, the level of fatigue, and the stress of the respondents.

This study focuses on the attitudinal aspects, and implicitly on the intangible aspects that contribute to entrepreneurial success, and consequently, the analysis does not aim to highlight exclusively tangible/hard-type components of entrepreneurial success such as: making a profit, the operational capacity of the head office, the number of employees and the ability to create new functions, the maintenance of medium and long-term financial liquidity, the ability to maintain and provide benefits to employees (e.g., logistics facilities), or the maintenance of long-term cooperation (more than a year) with customers.

The paper's findings cannot offer definitive responses on the presence of an absolute positive relationship between the motivation of achievement and entrepreneurship, but the approach is already derived from an acknowledged model in entrepreneurial studies. However, the received results are in the line with the GEM data and reports.

Even if respondents' perception of entrepreneurial success indicates macro-attitudinal interpretations, the average of positive perceptions of self-confidence, after a year of initiation and implementation of the start-up demonstrates widely, but not totally, the usefulness of support services provided by mentors. Indicators obtained from correlated variables [70] show the importance of acquiring a new understanding or obtaining entrepreneurial maturity, combined with newly gained skills, demonstrating the relevance of mentors' contribution and support teams for materializing a start-up [71]. The analyzed time segment, which refers to the transformative period of startup implementation, indicates the awareness of the need for additional training on the part of entrepreneurs, but also their ability to adapt to new business activity, considering further the constraints due to the COVID-19 pandemic crisis [12].

The study helped address the research gap regarding to what extent programs that finance start-ups from government sources contribute to the accumulation of personal and entrepreneurial capacities and provide an appropriate framework for the development and adoption of new solutions.

## 6. Conclusions

Despite the strong positive beliefs related to challenges, risk-taking, and commitment identified for new entrepreneurs, these do not decisively demonstrate a direct link with the quantitative variables of the business, such as creating jobs, higher income, and business self-reliability. In the study, however, there is a noticeable change in entrepreneurial attitude, based on subjective norms. Thus, for new entrepreneurs, the challenges such as fears regarding self-efficacy are revealed. There are responses from new entrepreneurs that indicate confidence in their ability to act independently and the ability to predict the evolution of their business in the short and medium-term.

In summary, 74 start-ups have achieved their intended objectives, to achieve turnover and maintain the number of employees during the implementation period. Concerning the positive perceptions of self-efficacy and indicators of entrepreneurial achievement at the end of the implementation period, we can assert that the analysis of the entrepreneurial transformation and the motivational perspectives of the ntrepreneurs suggest premises for the growth of the national economy.

Here it is noted that respondents over the age of 30 show more self-confidence in the ability to effectively manage the company without external support, and this can be justified by individual maturity which is likely a key factor in entrepreneurial success.

Finally, the study shows that the positive perception of entrepreneurial transformation demonstrates the objective awareness of one's entrepreneurial capacity. Consequently, in the process of entrepreneurial development, we find that the balance between "I want" and "I can" is better ensured, respectively, as the balance between intentions and the ability to be self-effective.

Thus, the scientific contribution of the presented study is to determine on the basis of economic and mathematical modeling the role of financial incentives for projects

along with the soft skills support to attract young entrepreneurs in Romania. Of course, the results obtained require confirmation through similar research in other countries, but it is obvious that the creation of such projects will be able to at least bring together active youth and increase their skills in social and entrepreneurial activities, to boost the entrepreneurial environment.

**Author Contributions:** Conceptualization, S.N. and V.G.; methodology, S.N. and V.G.; formal analysis, S.N., V.G. and A.S.; investigation, S.N., V.G., A.S. and G.K.; data curation, S.N., V.G. and G.K.; writing—original draft preparation, S.N. and V.G.; writing—review and editing, S.N., V.G. and G.K.; visualization, V.G.; supervision, S.N.; funding acquisition, S.N. All authors have read and agreed to the published version of the manuscript.

**Funding:** Project financed by Lucian Blaga University of Sibiu & Hasso Plattner Foundation research grants LBUS-IRG-2019-05.

**Institutional Review Board Statement:** Not applicable.

**Informed Consent Statement:** Informed consent was obtained from all subjects involved in the study.

**Data Availability Statement:** All data that was used for this research is available upon request.

**Acknowledgments:** The authors thank the 74 entrepreneurs who participated in the study and the implementation team of the project that granted funding and mentorship for the development of new businesses.

**Conflicts of Interest:** The authors declare no conflict of interest. The funders had no role in the design of the study; in the collection, analysis, or interpretation of data; in the writing of the manuscript, or in the decision to publish the results.

## Appendix A

**Table A1.** Correlation Matrix.

| | (TEA) | EBO | PO | PC | EEA | HSSE | EGCC |
|---|---|---|---|---|---|---|---|
| (TEA) | 1 | | | | | | |
| Established Business Ownership (EBO) | 0.87 (0.03) | 1 | | | | | |
| Perceived opportunities (PO) | 0.68 (0.04) | 0.39 (0.03) | 1 | | | | |
| Perceived capabilities (PC) | 0.87 (0.00) | 0.73 (0.01) | 0.66 (0.00) | 1 | | | |
| Entrepreneurial Employee Activity (EEA) | 0.70 (0.01) | 0.83 (0.02) | 0.08 (0.05) | 0.53 (0.03) | 1 | | |
| High Status to Successful Entrepreneurs (HSSE) | 0.88 (0.02) | 0.82 (0.03) | 0.58 (0.04) | 0.71 (0.02) | 0.63 (0.02) | 1 | |
| Entrepreneurship as a Good Career Choice (EGCC) | 0.86 (0.02) | 0.76 (0.04) | 0.53 (0.01) | 0.87 (0.01) | 0.75 (0.01) | 0.79 (0.04) | 1 |

Note: green colored—the column of results for the new entrepreneurs' indicator, orange-colored—the column of results for the mature one. The *p*-values are indicated per each correlation coefficient in the brackets. All coefficients can be accepted as significant at the 5%-level.

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
