# Peer review of "Fostering Entrepreneurial Ecosystems through the Stimulation and Mentorship of New Entrepreneurs"

_sustainability, doi:10.3390/su14137985_

Round 1

Reviewer 1 Report

The basic idea of the paper is very fine. Also the study design is good and use of literature is very strong. The sample is quite small for analysis, which can, however, be considered to be mostly descriptive by nature. The authors discuss the limitations and shortcomings in an advanced manner. 

Some detailed issues:

-row 202 it is mentioned that "correlation analysis is used to assess the impact of business conditions". Basically, correlation analysis can not be used to identify impacts, but relationships.

-Figures 3a, 3b and 4 could be very easily explained in the text, alternatively.   -in table 1, there is no need for 2-digit decimals. You should use "." instead of "," to separate the decimals

-figure 5 and table 1. There seems to be no additional information in figure 5 in relation to table 1

-it is good that the types of the businesses are discussed qualitatively

-also the discussion on limitations, especially as to data, is good  

Author Response

The comment of the Reviewer

The authors’ response

The basic idea of the paper is very fine. Also the study design is good and use of literature is very strong. The sample is quite small for analysis, which can, however, be considered to be mostly descriptive by nature. The authors discuss the limitations and shortcomings in an advanced manner. 

Thank you for the expert review and supportive comments and suggestions. That we fully accepted and corrected in the paper. All changes to the manuscript are marked up using the “Track

Changes” function.

-row 202 it is mentioned that "correlation analysis is used to assess the impact of business conditions". Basically, correlation analysis can not be used to identify impacts, but relationships

Fully agree, changed.

Figures 3a, 3b and 4 could be very easily explained in the text, alternatively.   -in table 1, there is no need for 2-digit decimals. You should use "." instead of "," to separate the decimals

Keeping the figures for illustration, we changed the explained sentences. Table 1 corrected.

-figure 5 and table 1. There seems to be no additional information in figure 5 in relation to table 1

Yes, Fig. 5 seems overwhelming with the Table 1 information, so we deleted it.

it is good that the types of the businesses are discussed qualitatively

Thank you

also the discussion on limitations, especially as to data, is good  

Thank you

Reviewer 2 Report

Dear colleagues!

Thank you for the opportunity to read the results of your research. I believe that they are of certain value in connection with the obvious, but often unconscious, need of beginning entrepreneurs to receive psychological and consulting support from experienced businessmen in building business.

At the same time, the study, in our view, has inherent limitations - experienced entrepreneurs may be disinterested in teaching and supporting aspiring entrepreneurs. Potentially, they can create competition in the future. And this demotivation is difficult to change, even through financial rewards. The authors should note how the study addresses this limitation.

The abstract and the introduction lack the purpose of the study and clearly stated novelty of the work. It is necessary to show the scientific increment to the available scientific results on this topic.

In the introduction, it is advisable to reflect only the main positions on the topic. The details should be moved to the literature review.

In the methods section, the authors denote the use of "correlation analysis is used to assess the impact of business conditions," but the results of correlation are unclear. Display the correlation matrix for clarity and comparison of the results of such analysis. This will show how you applied this analysis and its cumulative and comparative results. Indicate the values of the t-test to prove the significance of the correlation.

In the manuscript, it is useful to reflect how the experts for the Likert scale expert survey were selected.

I would also like to see a table of all the final results of the survey, particularly along the lines of improving entrepreneurial ability, perhaps in an appendix.

The authors note: "Calculating that 74 start-ups have achieved their intended objectives, to achieve turnover and maintain the number of employees during the implementation period." It is doubtful that all 74 startups were successful, as the authors point out.

The conclusions should more clearly reflect the scientific increment that the authors achieved in the process of researching this problem.

Author Response

The Reviewer’s comments

The authors' responses

experienced entrepreneurs may be disinterested in teaching and supporting aspiring entrepreneurs. Potentially, they can create competition in the future. And this demotivation is difficult to change, even through financial rewards. The authors should note how the study addresses this limitation

First of all, we would like to thank you deeply for the Reviewers’ supportive comments and suggestions that surely enhanced and polished the paper.

However, we do not see this position as a possible limitation.

It may seem that experienced entrepreneurs may not be interested in training and supporting start-ups, as they may create competition in the future. However, practice shows that such a warning is wrong. Many scientists note that with rising living standards, the expected level of wages ceases to play a decisive role, giving way to other factors. After obtaining a certain standard of living, entrepreneurs are not so much afraid of competition as interested in improving the well-being of others, facilitating their entrepreneurial environment. Not for nothing, most entrepreneurs, having received high incomes from their activities, begin to engage in charitable activities, volunteering, helping and mentoring of students. This is due to the desire to live in a successful society, where every successful person reduces the risk of crime, poverty, and other troubles, hence the risks for the businessman. For this reason, we can conclude that a significant proportion of successful entrepreneurs will be willing to join such projects, even for free or for a nominal fee. See examples in broadly held hackatons, start-up festivals, ideatons, free lectures for students.

The abstract and the introduction lack the purpose of the study and clearly stated novelty of the work. It is necessary to show the scientific increment to the available scientific results on this topic.

The goal is declared in the introduction: “This research has the goal to analyze how financial factors for establishing new ventures, multiplied by mentorship and similar networking and education factors, can impact new entrepreneurs and their perception of entrepreneurial self-efficacy” (lines 88-91).

The novelty is framed in the abstract.

The Introduction indicates 33 sources to highlight how the research can add to the existing scientific discourse and what could be its modest scientific increment.

In the introduction, it is advisable to reflect only the main positions on the topic. The details should be moved to the literature review.

Still, having the previous comment, we support the guidelines to keep 1-page introduction part. However, lit.review indeed disclose the main positions of the scientific results on the topic that we address and consider as the started point for our investigation.

In the methods section, the authors denote the use of "correlation analysis is used to assess the impact of business conditions," but the results of correlation are unclear. Display the correlation matrix for clarity and comparison of the results of such analysis. This will show how you applied this analysis and its cumulative and comparative results. Indicate the values of the t-test to prove the significance of the correlation.

Thank you for this comment. Actually, we decided not to present the correlation table as the lit.review on the topic is mostly descriptive and not deeply oriented on the econometric background, so for the sake of the potential readers we supposed not to put the table of the correlation matrix, but just indicate as we did – ie lines 246-254. However, if you insist we put it in Annex 1.

In the manuscript, it is useful to reflect how the experts for the Likert scale expert survey were selected.

Actually, we guess you mean experts for GEM data set. We did not overwhelm the paper with the methodology of GEM as it’s open-access information, and adding it could be considered as the plagiarism %. Here is the link to this information at the official web resource of Global Entrepreneurial Monitor: http://gem-consortium.ns-client.xyz/about/wiki

I would also like to see a table of all the final results of the survey, particularly along the lines of improving entrepreneurial ability, perhaps in an appendix.

Actually, the results release to what extent the entrepreneurs have adapted to the challenges encountered following the implementation of the business plan and after finalizing the sustainability period (18 months after establishing the new firm). The post-fact survey is presented, so the improving ability can not be traced in the form of the pre- and post-survey comparisons. Only, the final results and entrepreneurials’ perceptions and opinions are presented.

The authors note: "Calculating that 74 start-ups have achieved their intended objectives, to achieve turnover and maintain the number of employees during the implementation period." It is doubtful that all 74 startups were successful, as the authors point out.

We would like to highlight that it was the first project pool of start-ups, so we suppose that the most motivated were taken on board. So the success was quite forecasted. Our research was oriented on the more internal issues but not just hanging the mark “success / fail”. Quite obvious that while the project will be replicated more and more, the % of successful stories could be not 100%. But, the questions that were under our observation are not just narrowed to successful stories.

Round 2

Reviewer 2 Report

Dear Colleagues!

The manuscript has been somewhat improved, with the authors explaining a number of their positions. At the same time, some of the results obtained are of a probabilistic nature. Nevertheless, the study can be published. My advice to the authors is not to lose touch with aspiring entrepreneurs and after some time (a year to three years) to conduct another analysis of the success of their business. This will allow to connect the delayed level of stability and success of their business with self-assessment of the results of the current study.

I wish the authors good luck with further research!